# Corneal Penetration of Low-Dose Atropine Eye Drops

**DOI:** 10.3390/jcm10040588

**Published:** 2021-02-04

**Authors:** Henning Austermann, Frank Schaeffel, Ute Mathis, Verena Hund, Frank Mußhoff, Focke Ziemssen, Sven Schnichels

**Affiliations:** 1Center for Ophthalmology, Eberhard Karls University, 72076 Tübingen, Germany; henning.austermann@hotmail.de (H.A.); frank.schaeffel@uni-tuebingen.de (F.S.); ute.mathis@uni-tuebingen.de (U.M.); sven.schnichels@med.uni-tuebingen.de (S.S.); 2IOB Clinical Research Center, Myopia Research Group, CH-4031 Basel, Switzerland; 3Hospital Pharmacy, University Hospital Tübingen, 72076 Tübingen, Germany; verena.hund@med.uni-tuebingen.de; 4Forensic Toxicological Center GmbH, 80335 Munich, Germany; f.musshoff@ftc-muenchen.de

**Keywords:** low-dose atropine, myopia, ocular pharmacokinetics

## Abstract

Major studies demonstrating the inhibition of myopia in children and juveniles by low-dose atropine eye drops provide little information on the manufacturing process and the exact composition of the atropine dilutions. However, corneal penetration might significantly vary depending on preservatives, such as benzalkonium chloride (BAC), and the atropine concentration. Since there is a trade-off between side effects, stability, and optimal effects of atropine on myopia, it is important to gain better knowledge about intraocular atropine concentrations. We performed an ex vivo study to determine corneal penetration for different formulations. Atropine drops (0.01%) of different formulations were obtained from pharmacies and applied to the cornea of freshly enucleated pig eyes. After 10 min, a sample of aqueous humor was taken and atropine concentrations were determined after liquid–liquid extraction followed by high-performance liquid chromatography–tandem mass spectrometry (LC-MS/MS). The variability that originated from variations in applied drop size exceeded the differences between preserved and preservative-free formulations. The atropine concentration in the anterior chamber measured after 10 min was only 3.8 × 10^−8^ of its concentration in the applied eye drops, corresponding to 502.4 pM. Obviously, the preservative did not facilitate corneal penetration, at least ex vivo. In the aqueous humor of children’s eyes, similar concentrations, including higher variability, may be expected in the lower therapeutic window of pharmacodynamic action.

## 1. Introduction

According to meta-analyses, Cochrane reviews, and health technology assessments (HTAs), there is a large body of evidence that low-dose (0.01%) atropine is effective in reducing increases in childhood myopia [1,2,3,4,5]. The efficacy of the topical administration of atropine drops to control myopia has been known for almost 150 years [6]. In spite of their effectiveness in slowing axial elongation, the use of commercially available and higher-concentration atropine (0.1%, 0.5%, and 1%) is not well-accepted due to concentration-dependent side effects, including light sensitivity, reduced near vision, and a significant rebound effect after discontinuation. Therefore, off-label treatment with low-dose atropine is widely used in order to slow down the progression typically seen at school age.

Fine control of eye-length growth is largely achieved by image processing in the retina [7], without the need for central/cortical visual processing. The exact mechanism is still a subject of ongoing speculation and investigation [8,9]. In addition to the potential impact of accompanying substances, a very small amount of atropine broad-spectrum muscarinic acetylcholine receptor (mAChR) antagonist on the retina might be sufficient to cause regulatory changes [10]. Binding data and characteristics for muscarinic receptor antagonists have been published [11]. Dopamine (DA) and nitric oxide (NO) have been implicated in the process of myopia inhibition. However, some very potent mAChR antagonists (QNB (quinuclidinyl benzilate), dicyclomine, scopolamine, tropicamide) showed little effect in the chick model [12], and no link was identified between any mAChR genes and the risk of increased myopia [12].

Despite a consensus on basic efficacy, there is an ongoing debate about the duration and dosage of low-dose atropine [13,14]. A group from Rotterdam still promotes the use of 0.5% (or even 1%) atropine [15], while prescribing photochromatic multifocal glasses to reduce the photophobia secondary to dilated pupils. The large ATOM 2 study (400 children randomized in Singapore) reported a more favorable effect (less rebound, fewer side effects) of the lower dose [16]; however, a generally accepted standard of judging the effect primarily by the objective axis length has not yet been consistently achieved [17,18]. A recent randomized trial (LAMP study) reported the effective performance of other low concentrations (0.025%, 0.05%) of atropine eye drops over two years [19]. Although a concentration-dependent response seems reasonable [20], these data (a non-significant slowing of axial elongation for 0.01% atropine over 24 months) contradict previous studies and only reflect experience over two years. While other physicians have tested the daily use of a 0.02% concentration [21], 0.01% atropine is currently the most commonly used and investigated agent in the United States, East Asia, and Europe [22,23,24,25].

Unfortunately, most publications are very brief or do not specify exactly which formulations of atropine were used [16,19]. Most colleagues will use some sort of dilution of the 0.5% atropine eye drops used diagnostically. Dutch colleagues have indicated the ingredients of the drops they prescribe: benzalkonium chloride (BAC) as preservative, sodium edetate, boric acid, and purified water. The stability of atropine depends on the formulation [26]. Although the bioavailability and intraocular pharmacokinetics of atropine are well characterized [27], little is known about its precise penetration of the cornea. If we hypothetically assume a retinal concentration of 100 mM atropine (60 µM–6 mM), this would still be much greater than the affinity of atropine at the M4 receptor (K_d_ = 180 pM) [11]. With increasing levels, an increasing probability of off-target effects must also be assumed [28]. It has been speculated that, in addition to the retina, scleral fibroblasts may also be involved [29,30], or, more unlikely, the corneal endothelium [31]. It was identified early that pharmacodynamic effects also depend on iris pigmentation: mydriasis lasted significantly longer in pigmented animals (half-life 96 h vs. 43 h in pigmented vs. albino animals) [32]. Further studies have confirmed that that melanin binding of atropine caused prolonged action [33].

Especially considering the world-wide growth of prescriptions [14], we were interested in the impact of different formulations, the drop-to-drop variability, and the potential influence of preservatives such as BAC in the formulations. These questions should be investigated in simple, well-standardized ex vivo models to exclude other confounding factors as much as possible [34].

## 2. Materials and Methods

Freshly enucleated pig eyes were directly transferred to the laboratory on ice. Because they originated from a slaughterhouse (Emil Färber GmbH & Co. KG, Balingen, Germany), a separate animal experiment application was not required, in accordance with good laboratory practice. In an established set-up, the eyes were first checked for gross defects and then rinsed in a Tris-buffered saline (TBS). The globes were fixated with the cornea facing upwards. The eyes were moistened again with TBS buffer for about 30–45 s before application of the eye drops. We applied 40 µL of the drops to the cornea by pipette [35,36]. Drainage of the drops was not obstructed, resulting in additional contact of the drops with the sclera. This resembles a real-life situation and was performed in the same manner for all formulations. The eyes were received from the abattoir mostly without conjunctiva. For consistency, all external tissue was removed before drop application. After an incubation time of 10 min, a 200 µL sample of the aqueous humor was collected with a 1 mL Tbc syringe and a 30 G cannula. The samples were frozen at −80 °C for further analysis [37]. During a preliminary experiment, which corresponded to phase I (*n* = 6), the variability of the measurement method and the experimental setup was determined (±227 pg/mL). Based on an effect size of 1.4, the sample size was calculated to be 12 per group, using the two-sample mean calculator for mean difference to test the significance between the two groups (a = 0.05, 1−b = 0.9, 2-tailed). A Wilcoxon signed-rank test was used to compare distributions for non-normally distributed data. For data analysis, the statistical software package JMP V.14.2.0 (SAS Institute, Cary, NC, USA) was used.

### 2.1. Liquid Chromatography

Determination was performed after liquid–liquid extraction followed by high-performance liquid chromatography–tandem mass spectrometry (LC-MS/MS) [27]. The processing of the thawed samples included work-up of 50 µL of the aqueous humor sample with 50 µL 0.9% NaCl, 2 µL internal standard, and 100 µL buffer (5 mM ammonium formiate in water:methanol (85:15, *V*/*V*)). Potash lye was contained in the solvent (0; 0.05; 0.1; 0.25; 0.5; 1.25; 2.5; 5 ng/mL). The dilution factor was considered in the final result. After vortexing and centrifugation, 1 μL was directly injected into the LC-MS/MS system (Agilent Technologies 1290 Infinity with Sciex QTrap 6500, Agilent, Santa Clara, CA, USA). Separation was performed with a Kinetex 2.6 µm Biphenyl 100 Å, 50 mm × 2.1 mm (Phenomenex, Aschaffenburg, Germany), operating at 30 °C under gradient conditions for 7 min, with a flow rate of 0.5 mL/min, using eluents A (5 mM ammonium formiate in 0.1% formic acid in water) and B (5 mM ammonium formiate in 0.01% formic acid in methanol; positive electrospray ionization (5 kV), 555 °C, and MRM (multiple reaction monitoring) mode with three transitions). The gradient was as follows: it started at 5% B, followed by an increase to 75% B from 1.0 to 5.0 min. After rising to 100% B in 0.5 min, the gradient was held from 5.5 min to 6.5 min. Finally, gradient B was decreased to 5% at 7 min.

In accordance with good laboratory practice (GCP), the determination was made after regular calibration in the range of 0.05–100 ng/mL, with appropriate internal standards and controls. The lower limit of detection (LLOD) of the LC-MS/MS assay was 0.03 ng/mL, and the lower limit of quantification (LLOQ) 0.05 ng/mL. Precision was determined within 6.9%.

### 2.2. Atropine Preparations

The different preparations were obtained from pharmacies (Table 1). The pharmacies (Alte Apotheke, Stuttgart, Germany; Berg-Apotheke, Tecklenburg, Germany) did not know about the analysis. One unpreserved preparation corresponded to the study medication of an ongoing safety study (EUDRACT 2017-002588-17, Pharma Stulln, Stulln, Germany). Atropine POS 0.5% (0.2 mL; URSAPHARM, Saarbrücken, Germany) contains 5.0 mg of atropine sulphate and 0.50 mg BAC (benzalkonium chloride) per 10 mL, corresponding to 0.05 mg/mL. The approved medication is used to eliminate accommodation for diagnostic purposes (e.g., for pre-treatment for refraction determination or funduscopy; for penalization when occlusion treatment is not possible; to relieve accommodation spasms in hyperopia; in acute and chronic intraocular inflammations; in cases of injury to the iris or pupil; in ciliary body detachment; in cases of ciliolenticular block; or for disruption of synechiae).

In addition, atropine formulations of different concentrations were freshly prepared by the university pharmacy. The concentration of the preparations used was also controlled by means of LC-MS/MS. A balanced salt solution (BSS) served as a negative control without any atropine.

## 3. Results

Verification of the atropine concentration confirmed that the examined preparations contained a mean deviation of less than 12% from the declared concentration (95% CI (confidence interval): 92.5–113.2%). The standard error of the mean (SEM) was between 67.1 pg/mL and 119.7 pg/mL for all 0.01% formulations; the measurement variability was greatest for the 0.5% atropine (SEM 4.5 ng/mL, *n* = 12).

For three different formulations without preservatives, a concentration of atropine sulphate in the anterior chamber of 340.9 pg/mL was found (Figure 1; formulation 1: 322.3 pg/mL, formulation 2: 405.0 pg/mL, formulation 3: 300.7 pg/mL). Thus, the median proportion of the penetrated active ingredient detected in the aqueous humor 10 min after application was 1:3 × 10^7^. No atropine was measured in the control eyes.

The penetration of the preparations with preservative did not differ significantly from the preservative-free eye drops (Wilcoxon signed-rank test, *p* = 0.387). The mean concentration in the anterior chamber was 418.3 pg/mL for the three BAC formulations (284.8 pg/mL, 530.5 pg/mL, 429.4 pg/mL).

There was a clear dose-dependent increase of detected atropine sulphate in the aqueous humor (Figure 2). The mean level in the anterior chamber was 0.4 ng/mL (95%-CI: 299–459 pg/mL), 4.7 ng/mL (95%-CI: 2.8–6.7 ng/mL), and 14.6 ng/mL (95%-CI: 3.6–25.6 ng/mL) 10 min after administering 0.01%, 0.1%, and 0.5% eye drops. Although the fluctuations were clearly greatest with the highest concentration applied, the median percentage of the penetrated drug tended to be stable, 0.0000031%, 0.0000038%, and 0.0000017% of the actual quantity measured in the bottle. Compared to the nominally reported concentration, only a very small amount was found in the aqueous humor: 1:31115933 (dilution by 3.8 × 10^−8^ for 0.01%), 1:2136494 (dilution by 4.7 × 10^−7^ for 0.1%), and 1:104298 (dilution by 4.9 × 10^−7^ for 0.5%).

In the short follow-up period, the eyes showed no microscopically visible differences (stainability of the epithelium).

## 4. Discussion

While the majority of randomized trials (ATOM 2, LAMP) have not reported on the presence or precise concentrations of the preservatives/ingredients [16,19], recent trials provided specific measures within the study protocol, such as discarding opened drop vials after one week due to the dilution of BAC [38]. Only very few studies truly looked into chemical stability and microbiological safety, despite widespread use of the treatment [39,40]. Although our measurements do not allow any direct conclusions to be drawn about efficacy, nor therefore about the relevant minimum concentration that would be sensible to use, it can be stated that the presence of the preservative had only a negligible effect on the penetration of a single drop.

With regards to the pharmacokinetics after instillation, the tear concentration usually falls rapidly [41]; significant drug is transferred to the cornea, while the tear concentration remains higher than the corneal concentration. The cornea can serve as a storage depot for the aqueous humor. The rate of ocular drug movement is directly proportional to the concentration differential across the barrier changes; the passive diffusion of molecules across a non-saturated barrier generally adheres to first-order kinetics. Nevertheless, it is important to note that ultimately, bioavailability (which is the amount of the drug present at the desired receptor site) is critical.

The determination of drug stability is of major concern and has been well characterized in the case of atropine. In 1975, for the first time, in the United States Pharmacopeia (U.S.P.) XIX, a monograph on stability and stability testing was published. Today, the International Conference on Harmonization (ICH) guidelines Q1A through F are the harmonized guidelines regarding stability studies that require marketing authorization for the European, U.S., and Japanese drug markets [42]. These clearly define that stability testing is the responsibility of the manufacturer. For off-label therapy in particular, this responsibility is transferred to the pharmacist and the prescribing physician. ICH Guideline Q1A “Stability Testing of New Drug Substances and Medicinal Products” specifies the scope of stability testing for the approval of new active substances.

Atropine is the racemized form of (S)-hyoscyamine, which occurs naturally in nightshade plants such as mandrake, angel trumpet, jimson weed, and belladonna. Only the 1:1 mixture of (R)- and (S)-hyoscyamine is called atropine. As a tropic acid ester of tropyne, atropine is susceptible to pH-dependent hydrolysis catalyzed by both acids and bases. Kinetic studies have shown that the reaction is much slower in the acidic than in the basic environment, and therefore, the former plays only a minor role in the degradation of atropine sulfate in solution. The optimum stability of atropine with respect to hydrolysis to tropic acid is in the range of pH 3 to 4 [43]. However, the rate of dehydration to apoatropine is greatest in that range. Starting from apoatropine, dimerization in the manner of a Diels–Alder reaction to belladonnine as well as rehydration to atropine, can occur under extreme conditions, as can proton-catalyzed hydrolytic cleavage of apoatropine to atropic acid and tropanol. The hydrolysis products of atropine are not toxic, but do not possess anticholinergic activity [44]. Differences concerning the stability of atropine solutions stored in glass, plastic, or steel were found: regarding the shelf life of storing at ambient temperatures, atropine proved to be more stable in metal than in glass or plastic cartridges [45].

Informal manufacturer queries revealed that some of the dilution protocols differed significantly. Some added 0.0245 mL of a 1% BAC stock solution and 4.8755 mL of NaCl 0.9% to the commercial 0.5% preparation (leading to a pH of 4.65). The degree of acidity can increase the toxicity of BAC [46]. Later, a preparation of 0.2 mL 0.5% with edetate-containing BAC solution 0.1% (pH 4.6) and 0.49 g NaCl 0.9% and 10 mL was added to the national formulation collection (NRF 15.34, pH 4.83). A formulation containing boric acid, thiomersal stock solution, and hypromellose was originally chosen by one pharmacy because the preservative was more effective in the pH range [26].

### 4.1. Local Compatibility and Safety

Allergic conjunctivitis caused by atropine instillation has been reported [47,48,49]. Dutch authors relate the allergic reactions to the preservative contained (0.5%) [15]. In fact, they found more frequent reactions (6.8%) than described in other studies of the diluted formulations.

The wide distribution of the muscarinic receptors results in a wide range of biological effects. Cai et al. suggested an effective control of epiphora with transcutaneous application of atropine gel [50]. Atropine can reduce the oversecretion of the transplanted submandibular gland by modulating aquaporin-5 trafficking via hypersensitive mAChRs [51]. Therefore, topical instillation might also affect lacrimal gland secretion and alter the tear film on the ocular surface. The side effects of systemic atropine are dose-dependent (accommodation, dry mouth, and urinary retention). Psychological and central nervous effects are only expected for high doses (>10 mg) or intoxications. A paradoxical bradycardia has been described in the literature by some authors when administering intravenous doses of 0.5 mg. However, this was limited to an escalation in a very small sample of three subjects [52]. An earlier study in students observed a similar phenomenon for systemic doses of 0.25 mg [53].

In a comparison of topical ocular drugs, atropine is generally considered to be well tolerated by the corneal epithelium: preparations of up to 1% were found to cause no epithelial damage [54]. A comparative analysis showed that there was no increase in eye pressure in the group of treated children [55]. BAC can exhibit toxic effects on the tear film and the corneal epithelium [56,57]. Early on, it was suspected that such detergents could increase drug penetration [58]. However, our studies found no evidence that this seems to play a role in this context, presumably primarily because of the dilutions used. Although BAC was reported to produce a dose-dependent arrest of cell growth and death causing necrosis at higher concentrations [59], it is unlikely for lower concentrations.

The question remains as to what precise extent continuous therapy can contribute to the risk of dry eye or a stressed surface [47]. Apart from single reports of ocular discomfort, according to the ocular surface disease index (OSDI) and the objective parameters of the tear film at break-up time, meiboscore, or the anterior segment, optical coherence tomography was not changed at all in children treated for at least six months (*n* = 72) [60]. One study found no significant differences in the amount of tear secretion or signs of ocular surface inflammation between normal healthy children (*n* = 38) and myopic children receiving topical atropine treatment (*n* = 126) [61].

### 4.2. Mechanisms of Myopia Inhibition

Myopia is the most common developmental anomaly of the human eye. Although about half of high school graduates in Germany are myopic, with a further increase to 53% among students [62], there is no consistent evidence of a significant increase in each country and, in addition, no clear proof of digital media effects [63]. Myopia-associated ocular pathology can occur irrespective of the degree of myopia (without any cut-off or threshold); however, the prevalence of sight-threatening pathology, visual burden, and optical aberrations increases with higher degrees of myopia [64]. For each individual case, the clinical significance must be challenged, parents must be informed in detail, and critical judgment must be exercised [18]. Measuring and reporting axial length should be considered important, not only in future studies [65]. Previous studies have tried to find factors that might explain the differential response to atropine [66]. However, apart from the fact that the cohorts are usually far too small to allow a multifactorial analysis of all potentially relevant parameters, it is questionable whether all influencing factors can be taken into account so easily. Apart from the known importance of behavioral change (light exposure, near work) and genetic background, adherence, pigmentation of the iris, and axial length would have to be considered. There is not yet a precise consensus on when to discuss which intervention to use for a given child [3].

In chickens, glucagonergic amacrine cells have been found to increase the expression of the transcription factor ZENK during positive defocus (wearing positive lenses, focal plane in front of the retina), but to decrease expression during negative defocus (wearing negative lenses, focal plane behind the retina) [8]. Besides cholinergic antagonists (both nicotinic and muscarinic), several receptor agonists and antagonists are able to suppress myopia development within these models, including dopamine agonists (apomorphine), glucagon and its agonists, retinoic acid derivatives, VIP (vasoactive intestinal peptide), and adenosine antagonists. Inhibitors of apo-lipoprotein 2 and methylxanthine (MX-7) have been shown to be effective [8], even anti-VEGF (vascular endothelial growth factor) compounds inhibit the development of deprivation myopia in chickens [67]. Originally, it was thought that myopia development was inhibited by atropine by paralyzing accommodation. However, atropine also inhibits myopia development in chickens, which lack muscarinic receptors on the ciliary muscle and therefore can accommodate normally despite atropine administration [68]. Although speculation remains as to which is the most relevant muscarinic receptor, effects have been identified at the contralateral eye even in animal models, resulting from very low systemic concentrations [69]. In addition, several muscarinic antagonists have had no effect on myopia development in chickens, so even a non-muscarinic mechanism was suspected [12]. Carr and Stell have shown that an inhibitor of the nitric oxide (NO) synthase (L-NIO) abolishes the effect of atropine on myopia and have concluded that NO mediates the effect of atropine [70]. Another mechanism could be the interaction of atropine and dopamine release [8]. While proteomic studies in retinal cells are ongoing, treatment with atropine in humans was accompanied by an increase of choroidal thickness and abolished choroidal thinning due to hyperopic defocus [71]. This is reminiscent of very similar observations in animal models: muscarinic acetylcholine agonists cause choroidal thinning in chickens, whereas antagonists cause thickening [72]. Inhibition of eye length growth is typically preceded by choroidal thickening [73,74].

The limitations of an ex vivo model, which cannot represent factors such as tear flow or turnover of the tear film, must be considered. The drop size of the dispensers was not measured, as increasing drop size does not result in the penetration of more medication into the cornea [75]. In addition, longer times for corneal penetration (>30 min) have not yet been studied. However, the early examination time was to ensure that as few post-mortem changes as possible in the freshly enucleated porcine eyes (corneal epithelium, barrier disruption) had to be considered. Although species differences are unlikely given the structural homology of the cornea [34], the results cannot be extrapolated 1:1 to humans. After all, there are now numerous examples of other active ingredients whose penetration into the eye is not significantly affected by preservatives [76]. Last but not least, the size of the molecule and the concentration of the detergents are important [77]. The described variability of drug levels was found after administering similar drops (mean weight 40.3 mg, variance 0.254 mg). In clinical practice, the variability is certainly greater due to the dropper bottle and user.

## 5. Conclusions

Atropine 0.01% showed good, dose-dependent penetration into the cornea and anterior chamber, which was not significantly affected by additives and preservatives. Therefore, as long as long-term stability and microbiological safety are considered, appropriately unpreserved ophtioles and vials can also be used. Facing the additive effect of atropine with orthokeratology, the use of preservative-free solutions might allow a more gentle treatment, sparing the integrity of and avoiding toxicity to the corneal epithelium [38,78,79].

## Figures and Tables

**Figure 1 jcm-10-00588-f001:**
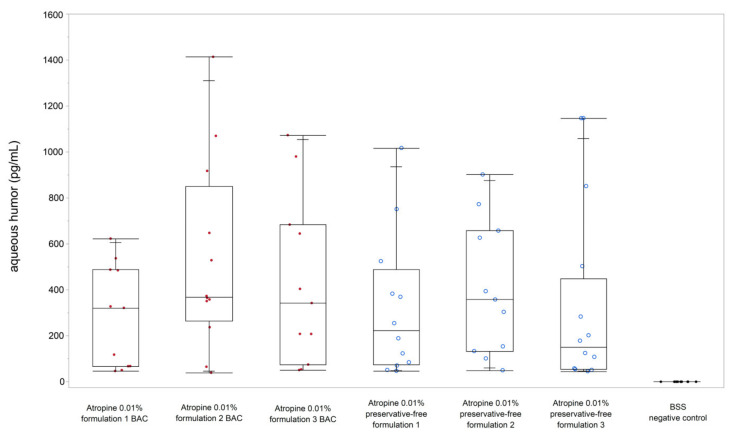
Concentration of atropine sulphate in aqueous humor for atropine 0.01% formulation with BAC (red) vs. preservative-free formulations. The bars within the box plots represent the median values, the edges represent the 25th/75th percentiles, and the whiskers extend from the ends of the boxes to the outermost data points within the quartile—1.5× of the interquartile range.

**Figure 2 jcm-10-00588-f002:**
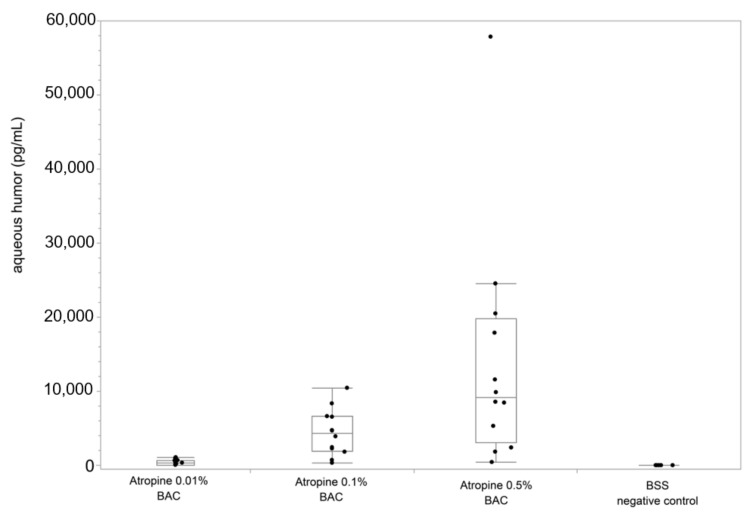
Concentration of atropine sulphate in aqueous humor for different atropine concentrations (0.01%, 0.1%, 0.5%, all including BAC). The bars within the box plots represent the medians, the edges represent the 25th/75th percentiles, and the whiskers extend from the ends of the box to the outermost data points within the quartile—1.5× of the interquartile range.

**Table 1 jcm-10-00588-t001:** Overview of tested formulations.

Formulation	Ingredients	Volume	Instructions for Use (Technical Information)	Shelf Life	Pharmacy
0.01% F1	Atropine sulphate 0.1 g/g, benzalkonium chloride 0.005%, sodium chloride, water	10 mL	Storage at 15–25 °C Can be used for 4 weeks after opening.	12 months	Bergapotheke
0.01% F2	Atropine sulphate 0.01%, benzalkonium chloride 0.005%, sodium chloride 0.9%, water for injection	10 mL	Keep protected from children Can be used for 4 weeks after opening.	8 months	Alte Apotheke
0.01% F3	Atropine sulphate 0.1 mg, sodium chloride, sodium edetate, preserved with 0.005% benzalkonium chloride	10 mL	Store below 25 °C Can be used for 4 weeks after opening.	8 months	University Pharmacy
0.01% PF1	Atropine sulphate 0.1 mg, sodium chloride, hydrochlorid acid, water for injection	0.5 mL	For single use. Discard the rest! Store at 15–25 °C.	12 months	Bergapotheke
0.01% PF2	Atropine sulphate 0.1 mg, sodium chloride, hydrochlorid acid, water for injection	10 mL	Store below 25 °C. Use 24 h after opening.	1 month	University Pharmacy
0.01% PF3	Atropine sulphate 0.1 mg n.d.	0.25 mL	Store in refrigerator 2–8 °C. After opening the sachet, the contents of the intact single-dose containers can be used for 1 month.	3 months	Study Medication
0.1%	Atropine sulphate 1 mg, sodium chloride, sodium edetate, preserved with 0.005% benzalkonium chloride	10 mL	Store below 25 °C Can be used for 4 weeks after opening.	8 months	University Pharmacy
0.5%	Atropine sulphate 5 mg/mL, benzalkonium chloride 0.05 mg/mL, sodium chloride, water	10 mL	Do not store above 25 °C. Use 4 weeks after opening.	18 months	Ursapharm

F, formulation, PF, preserved formulation, n.d., not defined.

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
