# Peer review of "Corneal Penetration of Low-Dose Atropine Eye Drops"

_jcm, 2021, doi:10.3390/jcm10040588_

Round 1
Reviewer 1 Report
The authors reported an interesting study about the penetration of 0.01 atropine eyedrop which might help readers.
They concluded that the atropine 0.01% (three drugs which is chosen by the authors) showed good, dose-dependent penetration into the cornea and anterior chamber. While it is good to know that the low dose Atropine can penetrate the cornea, the data is too simple for a whole paper.
The introduction is hard to understand. The discussion is not focused on the current data.
Author Response
Many thanks for your positive feedback to our draft. We can understand the concerns about what seem to be "simple" experiments at the first glance. Nevertheless, we would like to point out that the time expenditure and costs involved were not inconsiderable, having made every effort to ensure a clean methodology and careful scientific planning:
-
- Together with the Institute of Biometry, the pre-trials (case number planning according to phase 1) were planned to capture the scattering parameters.
- The sample size planning enabled a prospective study design, which allowed a cautious interpretation of valid results.
- During implementation, small details such as sample arrangement, consideration of blinded controls and allocation to multiple dates were observed to avoid the risk of systematic error.
- For comparison and analysis of preservation, our own internal controls (manufactured by the university pharmacy) were consulted in addition to preparations used from clinical supplies.
- The analysis of the original formulations, beyond the assayed aqueous humor, clarifies uncertainties/variations of the analytics as well as a relation to clinically relevant influencing factors (drop size, tear secretion, etc.).
Physicians and patients are very concerned and interested - especially given the off-label therapy that is manufactured from pharmacy to pharmacy - what concentrations are included and what intraocular levels are achieved. Low concentration poses particular challenges here, especially since intraocular bioavailability is influenced by additional factors such as iris pigmentation and efficacy may vary according to genetic influences on pharmacodynamics.
Therefore, this is a relevant knowledge gap that has not been addressed in the majority of high-level publications of randomized prospective trials. Although pathophysiological issues are not affected, the relation to binding affinities is relevant.
In the revised version, we have endeavored to make the introduction more comprehensible and to focus the discussion.

Reviewer 2 Report
This study is interesting insofar as it compares the corneal penetration of low doses of atropine eyedrops in several formulations.
Ex vivo models have been carried out, the administration has been standardized.The method used to quantify atropine is analysis technique is relevant.However, certain points remain to be completed and the discussion to refocus on the subject.
Methods
2.1.
Liquid chromatography: data is lacking regarding the extraction method (yield) and the precision of the analytical method (reproducibility, repeatability, accuracy). This information can be added as complementary data.
2.2 Atropine preparation: information on the different preparations is missing.
Make a table identifying each preparation (commercial and prepared by the university), all the excipients and their quantity
Results
The determination of the atropine concentration in the preparations exhibits a significant deviation from the theoretical concentration. How do you explain this variability ranging from 92.5% to 113.7% while a variation of less than 5% is expected for commercial or hospital preparations
What is the delay between preparation and administration? Under what conditions are the preparations stored? What is the shelf life of each preparation depending on the excipients and concentrations?
Title of figure 2 has been forgotten
Discussion
Revise the discussion to reorient it towards the focus of the manuscript.
Discuss the clinical interest of the concentrations obtained. Are they sufficient? Also discuss the sources of variability in preparations: analytical methods to quantify atropine? differences in formulations?
The medical devices used for administration (eyedrops) will probably be used be another source of variability? (in this study the volume , 40 µL) was taken with a pipette, which standardizes the administration. In real life, the volume of the drops emitted shows a greater variability.
Comment the impact of the formulation on corneal penetration. What are the parameters likely to improve the corneal diffusion of atropine? How can we adapt the formulation? what are the excipients likely to play a role? is there a buffer to regulate the pH?
What other preservatives can be used? Cetrimide?
The stability part of atropine presented in 4.1. is not justified at this level. This is a prerequisite for carrying out the preparations. If the formulations are not stable (eg formation of tropic acid) they should not be administered. The validation of the stability of the preparations studied must be mentioned in the introduction or in the methodology part. The chapter on antidotes and terrorism is irrelevant.
Author Response
This study is interesting insofar as it compares the corneal penetration of low doses of atropine eye drops in several formulations. Ex vivo models have been carried out, the administration has been standardized.
The method used to quantify atropine is analysis technique is relevant. However, certain points remain to be completed and the discussion to refocus on the subject.
- Many thanks for your positive feedback to our draft. We can understand the concerns about what seem to be "simple" experiments at the first glance. Nevertheless, we would like to point out that the time expenditure and costs involved were not inconsiderable, having made every effort to ensure a clean methodology and careful scientific planning:
We thank you for the fair evaluation and are happy about the good recommendations. Additions to the analytics and changes to the discussion unquestionably improve the quality of the manuscript.
Specific comments:
- Methods: 2.1.Liquid chromatography: data is lacking regarding the extraction method (yield) and the precision of the analytical method (reproducibility, repeatability, accuracy). This information can be added as complementary data.
- The requested information was supplemented. Especially the data on the calibration and precision of the methods allow well to evaluate the described measured values.
- Methods: Atropine preparation: information on the different preparations is missing
- The information from the manufacturers - if not previously available - was queried and supplemented. However, since no ingredients are analyzed, the information provided by the individual pharmacies cannot be independently confirmed.
- Make a table identifying each preparation (commercial and prepared by the university), all the excipients and their quantity
- The table was added. If requested, we could also provide images of each bottle.
4 .Results: The determination of the atropine concentration in the preparations exhibits a significant deviation from the theoretical concentration. How do you explain this variability ranging from 92.5% to 113.7% while a variation of less than 5% is expected for commercial or hospital preparations.
- We fully agree with the reviewer about the significant deviation of the measured concentration from the declared concentration. However, it must be taken into account that the fluctuations could also be amplified by the dilution steps required for the highly sensitive analysis.
- What is the delay between preparation and administration? Under what conditions are the preparations stored? What is the shelf life of each preparation depending on the excipients and concentrations?
- To exclude influences of stability in these studies, the unpreserved eye drops were tested immediately after preparation of the formulation. Unfortunately, we do not have details of all stability measurements, as each manufacturer is liable for the shelf life information itself and these data are not public. However, we are aware that in case adequate storage instructions are followed, unpreserved drops also show sufficient stability.
We believe the considerations are important and have therefore raised these points more clearly in the discussion.
- Title of figure 2 has been forgotten
- Please excuse the oversight. We were happy to add the title of the figure.
- Discussion: Revise the discussion to reorient it towards the focus of the manuscript.
- We must admit that some of the discussion of the original manuscript got a bit out of scope and left the focus of the experiments/results. Therefore, we gladly took up the suggestion and aligned the manuscript more stringently again.
- Discuss the clinical interest of the concentrations obtained. Are they sufficient? Also discuss the sources of variability in preparations: analytical methods to quantify atropine? differences in formulations?
- It is certainly reasonable to discuss the clinical relevance of the observations. Therefore, the significance of the concentrations and variations found have been picked up and interpreted with the necessary caution.
- The medical devices used for administration (eyedrops) will probably be used be another source of variability? (in this study the volume , 40 µL) was taken with a pipette, which standardizes the administration. In real life, the volume of the drops emitted shows a greater variability.
- In clinical practice, the variability is certainly greater. We have addressed the aspect of application (dropper bottle vs. pipette) more clearly in the revised version (page 16).
- Comment the impact of the formulation on corneal penetration. What are the parameters likely to improve the corneal diffusion of atropine? How can we adapt the formulation? What are the excipients likely to play a role? is there a buffer to regulate the pH? What other preservatives can be used? Cetrimide?
- It is a correct and good thought how the intraocular bioavailability could be optimized within the therapeutic range. Although we can of course - without appropriate data - say little about other preservatives, at least the presumed disruption and destabilization of the corneal barrier seems to play a minor role for BAC. Here, not least, the treated children likely differ from older glaucoma patients. This becomes more complex in accompaniment to contact lens correction.
In the revised version, we now addressed innovative models of drug delivery. Although in our experience the once-daily administration is conducive to good adherence, nano-formulations or depots, for example, would have the potential to allow less variability while still providing low concentrations with few side effects.
11.The stability part of atropine presented in 4.1. is not justified at this level. This is a prerequisite for carrying out the preparations. If the formulations are not stable (eg formation of tropic acid) they should not be administered. The validation of the stability of the preparations studied must be mentioned in the introduction or in the methodology part. The chapter on antidotes and terrorism is irrelevant.
- We have removed the paragraph on terrorism, poisons and antidotes. The information on stability has been added. We now limit ourselves to very briefly pointing out examples of unstable concentrations. However, it is completely correct that the penetration data do not provide any new information on stability, which requires repeated measurements over time.

Reviewer 3 Report
Dear Authors,
thank you for this interesting paper about atropin penetration into the eye. I like your thorough introduction and discussion in which you point out why this study is important. I agree with the limitations you mentioned.
My only question is: Since the eyes were enucleated, how did you manage the application on cornea and conjunctiva? Were parts of the sclera also in contact with the eye drops?
Kind regards!
Author Response
Thank you for the kind review and excellent comment: The exact experimental setup was presented in more detail to make the exact exposure (incl. limitations of the model) clearer.
There have been small parts of the sclera that were in short contact with the drops. In this experiment, rings or similar elements were deliberately omitted. The eyes were cleaned of fat or annex tissue but not remaining conjunctiva.
Round 2
Reviewer 1 Report
Thank you for your contribution.
Reviewer 2 Report
The manuscript was completed, the authors have followed the recommendations or argued when the elements were not available
Nouvel essai…